# Acute Changes in Serum Creatinine and Kinetic Glomerular Filtration Rate Estimation in Early Phase of Acute Pancreatitis

**DOI:** 10.3390/jcm11206159

**Published:** 2022-10-19

**Authors:** Paulina Dumnicka, Małgorzata Mazur-Laskowska, Piotr Ceranowicz, Mateusz Sporek, Witold Kolber, Joanna Tisończyk, Marek Kuźniewski, Barbara Maziarz, Beata Kuśnierz-Cabala

**Affiliations:** 1Department of Medical Diagnostics, Faculty of Pharmacy, Jagiellonian University Medical College, 30-688 Kraków, Poland; 2Diagnostics Department, University Hospital in Kraków, 30-688 Kraków, Poland; 3Department of Physiology, Faculty of Medicine, Jagiellonian University Medical College, 31-531 Kraków, Poland; 4Department of Anatomy, Faculty of Medicine, Jagiellonian University Medical College, 31-034 Kraków, Poland; 5Surgery Department, The District Hospital, 34-200 Sucha Beskidzka, Poland; 6Department of Surgery, Complex of Health Care Centers in Wadowice, 34-100 Wadowice, Poland; 7Chair and Department of Nephrology, Faculty of Medicine, Jagiellonian University Medical College, 30-688 Kraków, Poland; 8Department of Diagnostics, Chair of Clinical Biochemistry, Faculty of Medicine, Jagiellonian University Medical College, 31-066 Kraków, Poland; 9Chair of Medical Biochemistry, Faculty of Medicine, Jagiellonian University Medical College, 31-034 Kraków, Poland

**Keywords:** kinetic estimated glomerular filtration rate, acute pancreatitis, acute kidney injury

## Abstract

In patients with acutely changing kidney function, equations used to estimate glomerular filtration rate (eGFR) must be adjusted for dynamic changes in the concentrations of filtration markers (kinetic eGFR, KeGFR). The aim of our study was to evaluate serum creatinine-based KeGFR in patients in the early phase of acute pancreatitis (AP) as a marker of changing renal function and as a predictor of AP severity. We retrospectively calculated KeGFR on day 2 and 3 of the hospital stay in a group of 147 adult patients admitted within 24 h from the onset of AP symptoms and treated in two secondary-care hospitals. In 34 (23%) patients, changes in serum creatinine during days 1–3 of the hospital stay exceeded 26.5 µmol/L; KeGFR values almost completely differentiated those with increasing and decreasing serum creatinine (area under receiver operating characteristic curve, AUROC: 0.990 on day 3). In twelve (8%) patients, renal failure was diagnosed during the first three days of the hospital stay according to the modified Marshall scoring system, which was associated with significantly lower KeGFR values. KeGFR offered good diagnostic accuracy for renal failure (area under receiver operating characteristic—AUROC: 0.942 and 0.950 on days 2 and 3). Fourteen (10%) patients developed severe AP. KeGFR enabled prediction of severe AP with moderate diagnostic accuracy (AUROC: 0.788 and 0.769 on days 2 and 3), independently of age, sex, comorbidities and study center. Lower KeGFR values were significantly associated with mortality. Significant dynamic changes in renal function are common in the early phase of AP. KeGFR may be useful in the assessment of kidney function in AP and the prediction of AP severity.

## 1. Introduction

According to the 2012 revision of the Atlanta classification for acute pancreatitis (AP), organ failure including the cardiovascular system, lungs or kidneys is the main determinant of AP severity [1]. About one-third of patients with AP develop transient or persistent single or multiple organ failure, with renal failure being diagnosed in about 20% [2]. Moreover, the markers associated with renal function: serum creatinine [3] and blood urea nitrogen [4] are recognized predictors of severity of AP. Either serum creatinine or urea nitrogen is included in most prognostic scores currently used or proposed to evaluate AP severity such as Acute Physiology and Chronic Health Examination II (APACHE II), Bedside Index for Severity in AP (BISAP), Glasgow criteria, Ranson score, Harmless AP Score (HAPS), Japanese Severity Score (JSS) or Pancreatitis Outcome Prediction (POP) [5]. This reflects the fact that both underlying chronic kidney disease [6,7,8] and acute kidney injury (AKI) developing in the course of AP [9] are adversely associated with AP severity and mortality. 

The most widely accepted way of describing kidney function is through glomerular filtration rate (GFR), which use is supported by the clinical practice guidelines [10]. The assessment of GFR is based on measuring the renal clearance of exogenous (e.g., inulin, iohexol, iothalamate, or radiopharmaceuticals ^51^Cr-ethylenediaminetetraacetic acid and ^99m^Tc-diethylenetriaminepentaacetic acid) or endogenous substances (most commonly creatinine) [11,12]. Since such measurements are either invasive or cumbersome for patients (e.g., the assessment of creatinine clearance requires 24 h urine collection), routine clinical practice has widely adopted an estimation of GFR. Several formulas have been validated for this purpose, including those introduced by Chronic Kidney Disease–Epidemiology Collaboration (CKD-EPI), based on serum creatinine or cystatin C concentrations, age, sex and race [13,14]. The CKD-EPI formulas produce sufficiently adequate estimated GFR (eGFR) values in most cases when serum creatinine or cystatin C concentrations are stable over time, supporting clinical decisions in patients with chronic kidney disease [10]. However, these formulas do not work well when renal function is changing acutely, mainly because the changes in serum creatinine or cystatin C concentrations are slower (or appear with a lag time) compared to the changes in GFR [15,16]. Moreover, in patients with acute conditions, many non-renal factors (abnormalities in water balance, inflammation, altered metabolism and diet, or treatment including fluid resuscitation) influence the serum/plasma concentrations of creatinine and cystatin C [16]. In such patients, the measurement of GFR may be contraindicated (clearance of exogenous markers) or unreliable (creatinine clearance). Therefore, current diagnostic criteria for AKI are based on changes in serum creatinine over time, and not on measured GFR or eGFR [17].

To solve this problem, several authors [15,18,19,20] introduced the formulas enabling GFR estimation in a non-steady state, based on a general concept of creatinine clearance and incorporating the dynamically changing serum creatinine concentrations. Of those, the mathematically simplest formula of Chen [20] has been verified in several clinical studies involving patients of intensive care unit (ICU) [21] and renal transplant recipients [22,23]. In general, the clinical observational studies [21,22,23,24,25,26] supported the usefulness of KeGFR in the assessment of patients with acutely changing serum creatinine and suggested that KeGFR may improve drug dosing in such patients [27]. It has been postulated that evaluating GFR in patients with AKI would improve the assessment of patients in addition to laboratory markers of kidney tubular injury such as neutrophil gelatinase-associated lipocalin (NGAL), kidney injury molecule-1 (KIM-1), or so-called cell cycle arrest biomarkers: tissue inhibitor of matrix metalloproteinase-2 (TIMP-2) and insulin growth factor binding protein-7 (IGFBP-7) [28]. On the contrary, there are limited data showing a lack of agreement between KeGFR and measured GFR in patients with acutely changing kidney function [29,30].

We were not able to identify any studies using the kinetic eGFR (KeGFR) in patients or experimental animals with AP. Therefore, our aim was to estimate GFR in patients with AP using the KeGFR formula and to evaluate the diagnostic utility of KeGFR in the early stage of AP (first three days of hospital stay) to assess renal function and to predict the severity of AP. In the early phase of AP, hemodynamic changes are common and acute kidney injury may develop, resulting in dynamic changes in serum creatinine concentrations observed in clinical practice.

## 2. Materials and Methods

### 2.1. Patients and Definitions

The study was based on a retrospective analysis of data from two cohorts of adult patients with AP, recruited at the Surgery Department, District Hospital in Sucha Beskidzka, Poland, between January and December 2014 [31] and in the Department of Surgery, Complex of Health Care Centers in Wadowice, Poland, between March 2014 and December 2015 [32]. Both centers are secondary care regional hospitals. Both cohorts were recruited following the agreement of the appropriate bioethical commissions, and the present study protocol for the retrospective analysis of formerly obtained data was accepted by the Bioethical Committee of Jagiellonian University, Kraków, Poland (approval no 122.6120.241.2015 issued on 22 October 2015). The present study included patients for whom data were available to calculate KeGFR (i.e., serum creatinine was measured on each day of the study in known 24 h intervals).

The two cohorts included patients who were prospectively recruited based on consistent inclusion and exclusion criteria, namely, patients were included if they were admitted to hospital within the first 24 h following the onset of AP symptoms, they were diagnosed with AP based on the 2012 revised Atlanta classification (i.e., fulfilled at least two of three diagnostic criteria based on clinical, laboratory and imaging signs and symptoms) [1], they were adults (≥18 years old) and signed an informed consent for the study upon recruitment [31,32].

Patients with chronic pancreatitis, active cancer and chronic liver disease (viral hepatitis or liver cirrhosis) were excluded. Moreover, patients with a medical record of chronic kidney disease in whom baseline serum creatinine exceeded 170 µmol/L were excluded from the present study.

The final severity of AP was classified as mild (MAP), moderately severe (MSAP) or severe (SAP) according to the revised Atlanta classification [1] taking into account the persistent or transient cardiovascular, pulmonary, or renal failure as defined by modified Marshall scoring system (MMSS) [1], the systemic complications (exacerbation of comorbidities), and the local complications occurring during the entire hospital stay. Renal failure was defined according to MMSS [1] as serum creatinine exceeding 170 µmol/L.

The demographic and clinical data were collected including age, sex, preexisting comorbidities, BMI > 30 kg/m^2^, AP etiology, AP course and complications (presence of systemic inflammatory response syndrome—SIRS, pleural effusions, local complications, transient persistent organ failure as defined in revised 2012 Atlanta classification [1], mortality), and AP treatment (endoscopic retrograde cholangiopancreatography—ERCP, surgery, parenteral nutrition, length of hospital stay).

### 2.2. Laboratory Tests

In both cohorts, blood and urine samples were collected using the same timing, i.e., on admission (i.e., day 1) and then daily for two consecutive days of hospital stay (day 2 and day 3). Routine laboratory tests including complete blood count, biochemistry (serum albumin, glucose, urea, creatinine) and immunochemistry tests (serum C-reactive protein and plasma D-dimer) were performed on the day of blood collection in the laboratories associated with study centers. In both study centers, serum creatinine was assayed with the kinetic Jaffe method traceable to the isotope dilution mass spectrometry (IDMS) reference measurement procedure. Cobas 4000 analyzers (Roche Diagnostics, Mannheim, Germany) were used for the measurements. The reference intervals for serum creatinine were 44.0–80.0 µmol/L for adult women and 62.0–106.0 µmol/L for adult men, respectively. The serum urea/creatinine ratio (a unitless number) was calculated as serum urea expressed in mmol/L multiplied by 1000 (to unify the units) divided by serum creatinine in µmol/L.

Moreover, serum and urine samples were stored for additional laboratory tests that were performed in series following the collection of all samples. Briefly, blood and urine samples were centrifuged, and the supernatant was aliquoted, frozen and kept at −80 °C until analysis. The additional laboratory tests included serum cystatin C, β-trace protein (BTP), uromodulin, angiopoietin-2, soluble fms-like tyrosine kinase-1 (sFlt-1), neutrophil gelatinase-associated lipocalin (NGAL) and urine NGAL. The additional tests were not done on a part of the patients due to limited volume of samples. The numbers of patients with available results of the additional tests are presented in Appendix A (Table A1).

Serum cystatin C and BTP were measured using immunonephelometric method and Nephelometer II analyzer (Siemens Healthcare, Erlangen, Germany). The laboratory reference interval for cystatin C was 0.59–1.04 mg/L; the reference interval for serum BTP was <0.70 mg/L. The concentrations of sFlt-1 in serum were measured by electrochemiluminescence on Cobas 8000 analyzer (Roche Diagnostics, Mannheim, Germany). The concentrations of sFlt-1 in healthy subjects were 63–108 ng/mL. Urine NGAL was assessed with chemiluminescence microparticle immunoassay (Architect urine NGAL) on Abbott Architect analyzer (Abbott Laboratories, Chicago, IL, USA). The measurements were performed in the Diagnostic Department, University Hospital in Kraków, Poland.

Angiopoietin-2, uromodulin and NGAL in serum were measured with commercially available enzyme-linked immunosorbent assays (ELISA): Quantikine ELISA Human Angiopoietin-2 Immunoassay (R&D Systems, McKinley Place, MN, USA), Human Uromodulin ELISA and Human Lipocalin-2/NGAL ELISA (BioVendor, Brno, Czech Republic). The readings were performed with Automatic Micro ELISA Reader ELX 808 (BIO-TEK^®^ Instruments Inc., Winooski, VT, USA). According to the manufacturers of the kits, reference values in healthy adults were 1.065–8.907 ng/mL for angiopoietin-2, and 37.0–501.0 ng/mL for uromodulin, respectively. Serum NGAL concentrations in unselected adult donors were 14.4–169.2 ng/mL for men and 21.6–276.0 for women. The measurements were performed in the Department of Diagnostics, Chair of Clinical Biochemistry, Jagiellonian University Medical College, Kraków, Poland.

### 2.3. Equation Used to Estimate Kinetic GFR

KeGFR was estimated based on the formula presented by Chen [20], using eGFR based on serum creatinine calculated from 2021 CKD-EPI formula [13] as the substitute for creatinine clearance. Precisely, we used the following Formula (1): (1)$$\mathrm{KeGFR}=\frac{\min S_{Cr}\;\cdot \;\mathrm{eGFR}}{mean\;S_{Cr}}\;\cdot \left(1-\frac{24\;\cdot \;∆S_{Cr}}{∆t\cdot \;\max∆S_{Cr}}\right)$$
where min*S_Cr_* is minimum serum creatinine during the study, eGFR is the respective GFR calculated using 2021 CKD-EPI equation, and *mean S_Cr_* is the arithmetic mean of two consecutive serum creatinine results obtained during the study (e.g., on day 1 and day 2), Δ*S_Cr_* is a difference between the two consecutive serum creatinine results (e.g., day 2 creatinine–day 1 creatinine), Δ*t* is time in h between the two consecutive serum creatinine results (this equaled 24 h in our study), and maxΔ*S_Cr_* is a maximum possible rise in serum creatinine per day (such as may be observed in an anuric patient during AKI). For maxΔ*S_Cr_*, we used a value of 132.6 µmol/L (equal to 1.5 mg/dL as suggested by Chen [20] for adult patients). Since serum creatinine values are represented both in numerators and denominators in Equation (1), the equation works well irrespective of the unit used to express serum creatinine concentrations, and we used the values in µmol/L as reported by the study centers’ laboratories. Because the estimation of KeGFR requires two consecutive serum creatinine results, and we did not record pre-hospitalization serum creatinine values (these were unavailable in many patients), we were able to estimate two KeGFR values during the study: on day 2 (based on serum creatinine measured on admission, i.e., day 1 and on day 2), and day 3 (based on day 2 and day 3 serum creatinine results).

### 2.4. Statistical Analysis

Categorical data were summarized using number and percentage of the respective group and compared between groups using Pearson chi-squared test. Quantitative variables were summarized using mean and standard deviations if normally distributed and median and lower (Q1), upper quartile (Q3) if non-normally distributed. Since the vast majority of variables were non-normally distributed, Mann–Whitney test or Kruskal–Wallis test were used to compare the values between two or three groups, respectively. Spearman rank correlation coefficient was calculated to assess correlations between variables. Multiple linear and logistic regression was used to verify if the association between KeGFR and age is independent of the study center. The diagnostic accuracy was evaluated using receiver operating characteristic (ROC) curves. The cut-off values were selected at maximum Youden index. Multiple logistic regression served to assess KeGFR as a predictor of renal failure and SAP with adjustment for clinically significant confounders (age, sex, preexisting comorbidities, study center). Two-tailed statistical tests were used; the results were considered significant at *p* < 0.05. Statistica 13.3 (Tibco Software Inc., Tulsa, OK, USA) and associated Plus Bundle ver. 5.0.96 (StatSoft Polska, Kraków, Poland) were used for computations.

## 3. Results

### 3.1. Characteristics of Studied Patients with Acute Pancreatitis

The study included 147 patients (58 women and 89 men, between 19 and 86 years of age, with a median age of 56 years), of whom 81 (55%) were recruited in the Department of Surgery, Complex of Health Care Centers in Wadowice and 66 (45%) in the District Hospital in Sucha Beskidzka. Overall, there were 67 (46%) patients with MAP, 66 (45%) with MSAP and 14 (10%) with SAP (Table 1). The clinical characteristics of patients, including etiology, the course of AP and treatment needs differed according to severity (Table 1). The median length of hospital stay in the total cohort was 10 days (Q1; Q3: 6–15 days). Seven (5%) patients died: one death occurred during the first week of hospital stay (day 6) and six during the late stage of AP (days 10–31). According to MMSS (i.e., based on serum creatinine > 170 µmol/L), renal failure was diagnosed in 12 (8%) patients during the first three days of hospital stay, including 5 with MSAP and 7 with SAP (Table 1). None of the patients were subjected to renal replacement therapy during the study period. The study included five patients with preexisting chronic kidney disease in stage G1–G3a according to Kidney Disease: Improving Global Outcomes (KDIGO) [10]. Of them, two developed renal failure during the study as defined by MMSS.

The results of laboratory tests on admission (Table 2) reflected the severity of AP. The concentrations of most markers related to kidney function, i.e., serum urea, creatinine, cystatin C and NGAL, as well as urine NGAL, were significantly higher in patients with SAP. Conversely, there were no differences between MAP, MSAP and SAP patients regarding serum BTP and uromodulin. Serum concentrations of markers associated with endothelial dysfunction (angiopoietin-2 and sFlt-1) were highest in SAP (Table 2). 

### 3.2. KeGFR Values and Associations with Patients’ Characteristics

Two KeGFR values were calculated in each patient during the study period: day 2 KeGFR was estimated based on serum creatinine concentrations measured on admission and on day 2 of hospital stay, while day 3 KeGFR was based on serum creatinine concentrations measured on days 2 and 3. Because the information about serum creatinine measured before the admission was unavailable in the vast majority of patients, we were not able to estimate KeGFR values on the day of admission.

On both day 2 and 3 of the hospital stay, KeGFR values were strongly negatively associated with age (Table 3). The association was similar in patients recruited in both centers (Figure 1) and was independent on the site of recruitment in multiple linear regression. Moreover, in multiple logistic regression, the differences in KeGFR observed between the study centers (median 104 in Wadowice on both day 2 and 3 versus median 88 and 93 in Sucha Beskidzka on day 2 and 3; *p* < 0.001) were fully explained by the difference in patients’ age (median 46: Q1; Q3: 36; 61 years versus median 63; Q1; Q3: 44; 78 years; *p* < 0.001; Appendix A, Table A2). 

KeGFR values in men (median 104; Q1; Q3: 87; 117 mL/min/1.73 m^2^ on day 2 and median 104; Q1; Q3: 91; 116 mL/min/1.73 m^2^ on day 3) were significantly higher than in women (median 89; Q1; Q3: 73; 104 mL/min/1.73 m^2^ on day 2; *p* = 0.004 and median 89; Q1; Q3: 80; 104 mL/min/1.73 m^2^ on day 3; *p* = 0.003).

KeGFR values correlated well with studied markers of glomerular filtration, i.e., serum cystatin C and BTP in addition to serum creatinine (Table 3). Moreover, the correlations between KeGFR and serum urea were fairly strong. On the contrary, KeGFR did not correlate with urine NGAL and only weakly correlated with serum NGAL and uromodulin, the markers associated with tubular injury or tubular function. Furthermore, a weak correlation was observed between KeGFR and serum urea/creatinine ratio. Moreover, we observed weak to moderate negative correlations between KeGFR and the markers of endothelial injury: serum angiopoietin-2 and sFlt-1 (Table 3). 

### 3.3. KeGFR in Assessment of Kidney Function in the Early Phase of AP

In 12 patients with renal failure diagnosed according to MMSS, KeGFR values were significantly lower as compared to patients without renal failure and enabled the diagnosis of renal failure with high accuracy (Figure 2; Table 4). The areas under the ROC curves (AUROC) did not differ significantly between KeGFR (Table 4) and serum creatinine (0.983 on day 1; 0.970 on day 2 and 0.990 on day 3; Figure 2C,D). This was despite the fact that serum creatinine was used to diagnose renal failure as defined by MMSS. Slightly lower AUROC values observed for KeGFR may be associated with the dynamic changes in renal function observed during the study in the 12 patients with renal failure. In seven (58%) of them, the highest creatinine was recorded on admission, and the concentrations decreased on day 2 or 3 in response to therapy, which was reflected by increasing KeGFR values. In the remaining five patients with renal failure, the highest serum creatinine during the study was noted on days 2 or 3.

The KeGFR cut-off values to diagnose renal failure selected at the maximum Youden index were 35.5 mL/min/1.73 m^2^ on day 2 (allowing for 75% sensitivity and nearly 100% specificity) and 86.2 mL/min/1.73 m^2^ on day 3 (allowing for 100% sensitivity and nearly 80% specificity). In Table 4, we additionally report the cut-off values selected at the second maximum Youden index, to enable a better comparison of day 2 and day 3 KeGFR. 

In multiple logistic regression analysis, lower KeGFR on both day 2 and day 3 of hospital stay was significantly associated with renal failure, independently of age, sex, preexisting comorbidities, a final diagnosis of SAP and study center (Table 5). When preexisting chronic kidney disease was included in the models instead of preexisting comorbidities, the association remained significant: odds ratios for renal failure equaled 0.93 (0.89–0.97); *p* < 0.001 for day 2 KeGFR, and 0.87 (0.79–0.94); *p* = 0.001 for day 3 KeGFR.

In 34 (23%) of the studied 147 patients with AP, including 3 (4%) with MAP, 21 (32%) with MSAP and 10 (72%) with SAP (*p* < 0.001; significant difference between all groups in post hoc tests), serum creatinine was changing acutely in the first three days of the hospital stay and the absolute difference between two serum creatinine concentrations obtained within 48 h (i.e., between the day of admission and day 3 of the hospital stay) exceeded 26.5 µmol/L, i.e., a value consistent with the KDIGO definition of AKI [17]. In eight (24%) of these patients, serum creatinine increased from day 1 to day 3 of the hospital stay while in the remaining 26 patients, day 3 serum creatinine was lower than the day 1 value (Figure 3). The KeGFR values on day 2 and 3 differed significantly between the two subgroups (Figure 3E,F), allowing for nearly complete discrimination on day 3 (AUROC 0.904 on day 2 and 0.990 on day 3). The median value of day 2 KeGFR was 4.6 times higher and the median value of day 3 KeGFR was 4.8 times higher in the subgroup with ≥26.5 µmol/L serum creatinine increase compared to the subgroup with ≥26.5 µmol/L serum creatinine decrease during days 1–3 of the study (the respective medians of serum creatinine were 3 times and 3.3 times lower). Figure 3A–D presents the changes in serum creatinine and the respective KeGFR changes in individual patients over the study period.

### 3.4. KeGFR as a Predictor of AP Severity

In patients with SAP, average KeGFR values were significantly lower as compared to those with MAP and MSAP (Figure 4), while they did not differ significantly between patients with MAP and MSAP. On both day 2 and day 3 of the hospital stay, low KeGFR values predicted SAP with moderate diagnostic accuracy (Figure 4C; Table 6); although, the AUROC values did not differ significantly from those obtained for serum creatinine (i.e., 0.766 on day 1; 0.757 on day 2; and 0.744 on day 3, respectively; Figure 4D). 

In multiple logistic regression, lower KeGFR values on day 2 and on day 3 significantly predicted SAP independently of age, sex, preexisting comorbidities and the recruitment center (Table 7). When the analysis was performed excluding patients diagnosed with renal failure, day 2 KeGFR (odds ratio: 0.98; 95% confidence interval: 0.92–0.99; *p* = 0.023) was still a significant predictor of SAP; although, day 3 KeGFR was not (odds ratio: 0.95; 95% confidence interval: 0.90–1.01; *p* = 0.08).

Moreover, lower KeGFR values were significantly associated with AP mortality in univariate logistic regression (OR per 1 mL/min/1.73 m^2^ increase in KeGFR was 0.97; 95% CI: 0.95–0.99; *p* = 0.009 on day 2 and OR = 0.97; 95% CI: 0.95–0.99; *p* = 0.006 on day 3, respectively). After adjustment for confounders (age, sex, preexisting comorbidities and study center), this association remained significant on day 2 (OR = 0.97; 95% CI: 0.95–0.99; *p* = 0.043) but became non-significant on day 3.

## 4. Discussion

In our study, significant acute changes in serum creatinine were observed in nearly one-fourth of patients with AP during the first three days from admission. KeGFR was an accurate marker of renal failure defined according to MMSS in the early phase of AP. Although the diagnostic accuracy for renal failure was similar for KeGFR and single serum creatinine measurements on days 2 and 3 of the hospital stay, the changes in KeGFR associated with the dynamic changes in kidney function were more pronounced than the respective changes in serum creatinine. Moreover, KeGFR values assessed in patients with AP during the initial days of the hospital stay significantly predicted a severe course of the disease and mortality. 

To the best of our knowledge, KeGFR has not been studied before in patients with AP, either in the context of kidney injury or as a marker of severity. In the initial phase of AP, dynamic changes in serum creatinine, indicating the dynamic changes in glomerular filtration, are common. Therefore, steady state eGFR would not adequately reflect kidney function in many patients with early stage of AP. In 23% of our patients, the difference in two serum creatinine concentrations obtained within the first three days of hospital stay (i.e., within 48 h) exceeded 26.5 µmol/L. Such a difference has been recognized by KDIGO guidelines [17] as indicative of AKI, and AKI diagnosed with the KDIGO definition is associated with increased mortality [33,34]. In the present study, KeGFR correlated negatively with the markers of endothelial injury (angiopoietin 2 and sFlt-1), in line with our previous observations showing the association between these endothelial markers and early kidney injury in AP [31,35]. This supports the pathophysiological role of endothelial dysfunction in early AP, associated with increased vascular leakage and hemodynamic changes that also influence glomerular filtration.

In the studied group, KeGFR was strongly negatively correlated with age, which may reflect both the well-known decrease in baseline GFR observed in elderly patients [10] and the more pronounced hemodynamic changes in older patients or the higher risk for AKI in older patients with AP [2]. Moreover, although we calculated KeGFR based on body surface area-standardized eGFR values, women were characterized with lower KeGFR than men. Both the baseline patients’ characteristics were therefore included as confounders in the multiple regression analyses.

We observed significant changes in serum creatinine mainly in patients with MSAP and SAP. Moreover, renal failure diagnosed in line with MMSS was most prevalent in SAP. Consequently, lower KeGFR was significantly associated with the diagnosis of SAP and mortality. In 2021, Tod et al. [36] published an analysis of data from a Hungarian AP registry. They reported a significant association between lower steady-state eGFR within 24 h from admission and the diagnosis of SAP, increased length of hospital stay, and mortality [36]. We did not observe a significant correlation between KeGFR and length of hospital stay; nonetheless, we observed lower KeGFR values in patients with SAP (even after excluding those diagnosed with renal failure) and a significant association between low KeGFR and mortality. We must, however, underlie several doubts regarding the report of Tod et al. [36]: first, their method of eGFR calculation is unclear (although they stated that they used the 2009 CKD-EPI equation based on serum creatinine, the equation shown in their report differs significantly from the 2009 CKD-EPI equation); further, the number of patients included in the analysis is also unclear (they claimed to include data of 1224 patients; however, they apparently excluded 511 patients with eGFR > 90 mL/min/1.73 m^2^).

In addition to serum creatinine, we measured other markers of glomerular filtration (cystatin C, BTP) in available serum samples of our patients. Since these results were not available in the whole studied group, we did not calculate KeGFR based on serum cystatin C or BTP. It is, however, possible to use Chen’s equation with non-creatinine filtration markers; for example, the study of Pianta et al. [23] showed the clinical utility of KeGFR based on plasma cystatin C in renal transplant recipients. We have previously evaluated serum cystatin C and BTP as markers of AKI in the early phase of AP, showing the moderate diagnostic utility of both markers [37]. Now, we observed strong negative correlations between creatinine-based KeGFR and serum concentrations of cystatin C and BTP. In contrast, the correlations between KeGFR and serum NGAL were weak and there were no significant correlations between KeGFR and urine NGAL in our present study. Since our previous results suggest that urine NGAL is a good diagnostic marker of AKI in the early phase of AP [37], we may hypothesize that the simultaneous use of KeGFR with urine NGAL (or alternatively, other tubular markers) could be a sensitive diagnostic strategy for renal injury and dysfunction.

Although the idea of GFR estimation in a non-steady state is not new, KeGFR has regained interest in recent years, following the work of Sheldon Chen [20], who proposed a relatively simple equation allowing for the estimation of GFR from the results of repeated serum creatinine measurements performed in patients with acutely changing kidney function. The idea behind the equation proposed by Chen was clearly described in his article [20] and the calculations were easy to adopt in clinical research. This equation was used to calculate KeGFR in observational clinical studies including patients treated in ICU [21,24,25], general hospital patients [26] and renal transplant recipients [22,23]; the same equation was used in our study. There are, however, some discrepancies in the use of the equation between the studies. Namely, the maximum daily change in serum creatinine has either been estimated based on a chosen creatinine concentration, respective eGFR and the volume of distribution for serum creatinine (e.g., by Pianta et al. [23] and O’Sullivan and Doyle [24]), or substituted by a constant number (133 µmol/day) as originally suggested by Chen [20] (e.g., in studies [21,22] and ours). Moreover, the previous studies used either the Modification of Diet in Renal Diseases (MDRD) [24] or the 2009 CKD-EPI equation [21,22,23] as the basic estimation of GFR, whereas we decided on the 2021 race-independent CKD-EPI equation [13]. The 2009 CKD-EPI equation has been shown to better estimate measured GFR than MDRD and has been recommended by KDIGO guidelines to evaluate patients with chronic kidney disease [10,38]. Nonetheless, the two (2009 and 2021) CKD-EPI equations give similar numbers in Caucasian adults and the 2021 equation has been shown to be slightly more consistent with measured GFR in this population [39]. Further discrepancies between the studies regard serum creatinine measurements. Pianta et al. [23] measured serum creatinine with the enzymatic method while O’Sullivan and Doyle [24] and Dash et al. [22] used the kinetic Jaffe method, similar to our study. 

Irrespective of the differences, the results of clinical observational studies [21,22,23,24,25,26] support the clinical utility of KeGFR in various patient populations with acutely changing serum creatinine and are in general agreement with our results. In 2015, Dewitte et al. [21] published an analysis of 57 patients who developed AKI within 24 h of admission to ICU. They calculated KeGFR on the day following admission and measured urine NGAL, TIMP-2 and IGFBP-7 on admission and on the following day. They assessed the diagnostic accuracy of the studied markers for two distinct outcomes: renal recovery after 48 h in response to fluid resuscitation, and major adverse kidney events (death, renal replacement therapy or doubled serum creatinine at 30 days from admission). KeGFR was an accurate predictor of early recovery from AKI, with an AUROC of 0.87. Both the product of TIMP-2 and IGFBP-7 concentrations and KeGFR showed similarly high diagnostic accuracy for major adverse kidney events (AUROC of 0.80) [21]. The study [21] included only two patients with AP (3.5% of the studied group); however, the primary end-point of early (within 48 h) recovery from AKI following fluid resuscitation corresponds well to the definition of transient organ failure in AP, i.e., resolving within 48 h (in contrast to persistent organ failure in SAP). In 2017, O’Sullivan and Doyle [24] retrospectively compared KeGFR and MDRD-based eGFR as predictors of AKI, renal replacement therapy and the mortality of 107 patients admitted to high dependency and intensive care units of a district hospital. AP was not mentioned among the patients’ diagnoses. KeGFR calculated using two initial serum creatinine concentrations measured after admission predicted AKI and renal replacement therapy with better diagnostic accuracy than eGFR calculated with the MDRD equation at admission. However, neither KeGFR nor eGFR (MDRD) predicted 30-day mortality [24]. De Oliveira Marques et al. [25] retrospectively analyzed data from a large cohort (over 13,000) of patients during the first week in ICU. The lowest KeGFR recorded during the first week of stay in the ICU was shown to be associated with the outcome variables (need for renal replacement therapy, hospital mortality and 1-year mortality) in addition to the maximum AKI stage as defined by KDIGO guidelines [17]. There was moderate concordance between the renal function assessed by KeGFR and AKI diagnosis and stages as defined by the KDIGO guidelines. Christiadi et al. [26] evaluated KeGFR in 140 hospital patients with positive electronic AKI alert (i.e., serum creatinine results fulfilling the KDIGO [17] definition of AKI) and 242 control patients without AKI. The decrease in KeGFR of 8–10% or more was identified as an accurate marker of AKI. The authors [26] observed that decreasing KeGFR may allow for 24 h earlier identification of patients with AKI compared with a creatinine increase fulfilling KDIGO criteria. All these results are consistent regarding the clinical utility of KeGFR in the detection of acute decrease in renal function and related increase in adverse clinical outcomes in patients admitted to hospital, high dependency or intensive care units. Given the lack of previous studies on KeGFR in patients with AP, these patients’ populations are most comparable to ours. 

Clinically, KeGFR enables the translation of changes in serum creatinine in individual patients into more meaningful numbers [24,27]. GFR may better guide therapeutic decisions than serum creatinine alone. Calculating KeGFR allows widely accepted GFR thresholds to be used to initiate renal replacement therapy or to adjust doses of therapeutic drugs in patients with changing renal function where steady-state eGFR equations are inappropriate. As shown by Kwong et al. [27], in critically ill patients, the use of KeGFR instead of eGFR would result in significant changes in drug dosing in about 25% of patients. 

Scarce data are available on KeGFR validation against measured GFR in patients with dynamically changing kidney function; we were able to identify two such studies, published in 2020 and 2021. Both studies used iohexol clearance as the reference standard of measured GFR. Sangla et al. [29] included 66 adult patients admitted to general ICU. They compared Chen’s KeGFR and CKD-EPI eGFR based on serum creatinine and cystatin C with measured GFR. All studied equations overestimated GFR. The estimations based on serum creatinine led to worse results over time, which appeared to be associated with the development of sarcopenia leading to lower production of creatinine [29]. Desgrouas et al. [30] compared eleven eGFR equations with measured GFR in 57 intensive care patients with shock. Measured GFR on day 1 of the study was significantly lower in most patients when compared to baseline steady-state eGFR, reflecting quickly decreasing renal function. All studied equations resulted in a significant positive bias (between 12 and 46 mL/min/1.73 m^2^) versus measured GFR. In fact, Chen’s KeGFR was shown to be less consistent with measured GFR than several other formulas (e.g., Yashiro [19] or Moran and Myers [18]) [30]. Moreover, in both studies [29,30], all formulas used to estimate GFR resulted in unacceptable imprecision. 

In the present study, we chose the simplest KeGFR formula proposed by Chen. However, his further works suggested more complicated formulas allowing for theoretical improvements in KeGFR estimation [40,41]. Although these formulas offer better estimates of KeGFR based on the dynamic modeling of creatinine distribution and elimination, their clinical use is hindered by the difficulties in mathematical calculations. The uptake of these formulas would possibly require a wide access to software designed to calculate KeGFR, verified in clinical studies. Optimally, KeGFR calculated using various proposed formulas should be verified against measured GFR in patients with acutely changing (decreasing or increasing) kidney function. 

## 5. Conclusions

Our study is the first assessment of KeGFR in patients in the early phase of AP. Our results confirmed that the dynamic changes in kidney function as reflected by acute changes in serum creatinine concentrations are common in patients admitted to hospital with AP and are mainly observed in those with moderate to severe disease. Such acute changes in serum creatinine preclude the use of steady-state eGFR equations for the assessment of kidney function in this group of patients. Although limited by a retrospective design, our study indicates that the KeGFR equation may offer a reliable clinical alternative for the assessment and monitoring of kidney function in the early AP. In the studied group, KeGFR allowed for high diagnostic accuracy for renal failure and moderate prognostic accuracy for SAP; however, it must be acknowledged that the diagnostic accuracy of KeGFR did not differ significantly from serum creatinine. 

Our study has several limitations. Most importantly, this was a retrospective analysis of previously acquired data. We were not able to perform additional laboratory tests on our patients; therefore, some information is available only in a part of the studied group. Moreover, because of the retrospective design, we were not able to assess KeGFR as a predictor of the further course of renal failure in the patients; we could only assess its diagnostic accuracy. The sample size was limited while the small number of patients with adverse outcomes (renal failure, severe AP or death) was partially a result of the inclusion of secondary care patients and should also be mentioned. Therefore, further studies are necessary to confirm our results in prospective studies in larger groups of patients with AP including more severe patients, designed to follow the kidney function over longer periods of time. Moreover, there is a need to compare the various ways of kinetic GFR estimation (various equations, various filtration markers) with measured GFR in patients with acutely changing kidney function. 

## Figures and Tables

**Figure 1 jcm-11-06159-f001:**
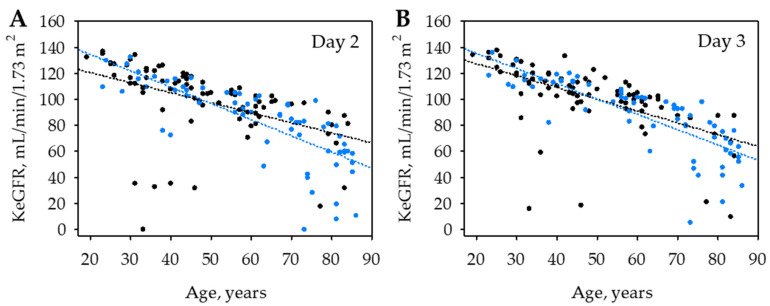
Association between kinetic estimated glomerular filtration rate (KeGFR) values on day 2 (**A**) and 3 (**B**) of hospital stay in patients with AP recruited in two study centers: the Department of Surgery, Complex of Health Care Centers in Wadowice, Poland (in black) and the District Hospital in Sucha Beskidzka, Poland (in blue).

**Figure 2 jcm-11-06159-f002:**
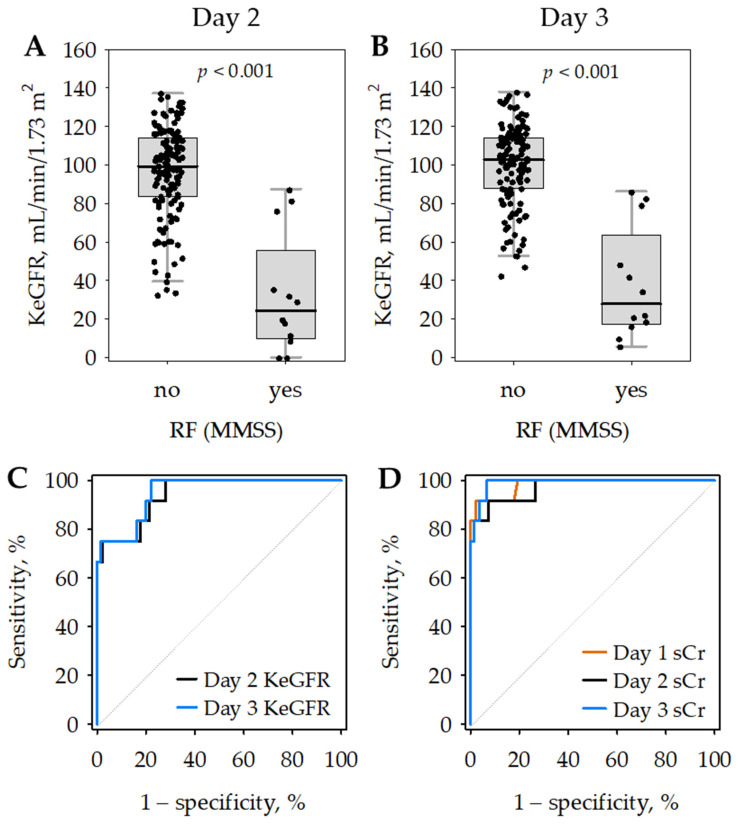
Kinetic estimated glomerular filtration rate (KeGFR) values on day 2 (**A**) and day 3 (**B**) of hospital stay according to the diagnosis of renal failure (RF) on days 1 to 3 based on the modified Marshall scoring system (MMSS). Data are shown as median (central line), interquartile range (box), non-outlier range (whiskers) and raw data (points). Receiver operating characteristic (ROC) curves showing the diagnostic accuracy of day 2 (black) and day 3 (blue) KeGFR for renal failure defined according to MMSS (**C**). For comparison, ROC curves for serum creatinine (sCr) as a marker of renal failure are presented on panel (**D**) (day 1—orange; day 2—black; day 3—blue).

**Figure 3 jcm-11-06159-f003:**
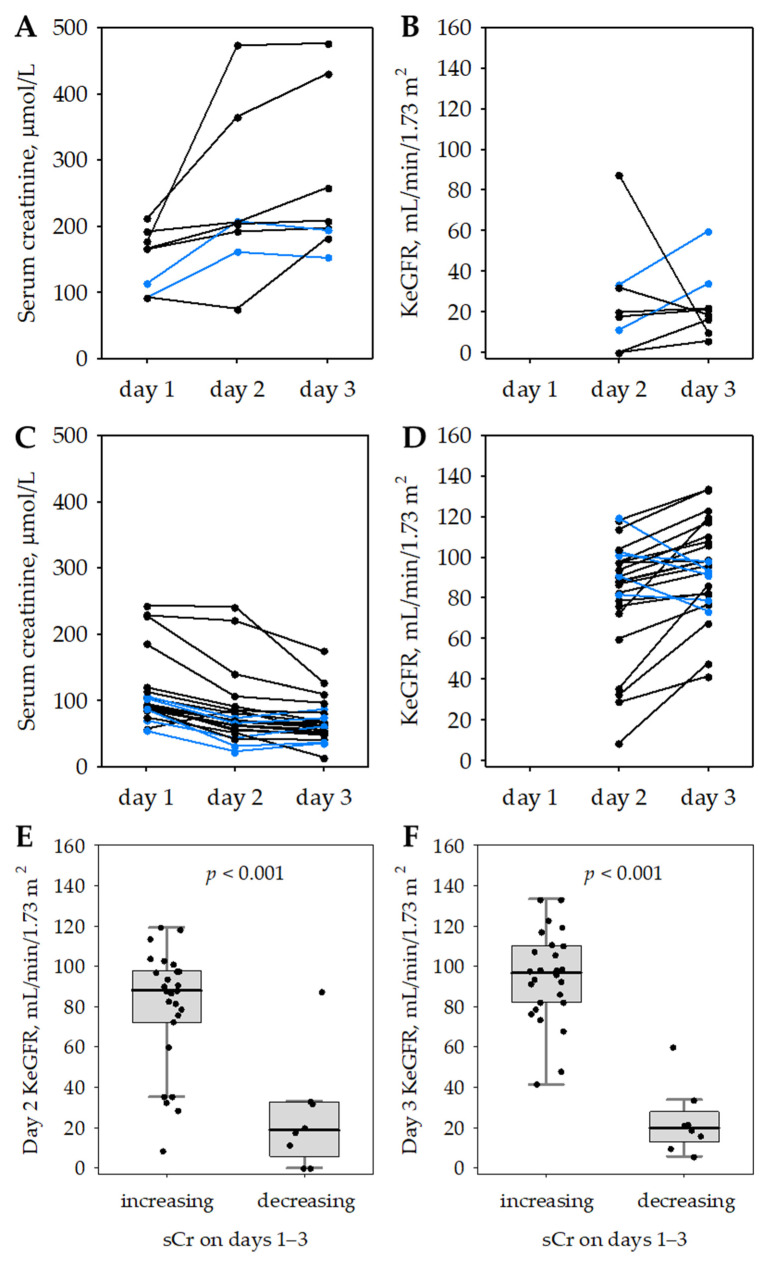
Longitudinal changes in serum creatinine (**A**,**C**), and KeGFR (**B**,**D**) in a subgroup of studied AP patients in whom serum creatinine increased by ≥26.5 µmol/L (**A**,**B**) or decreased by ≥26.5 µmol/L (**C**,**D**) during the first three days of hospital stay. The blue lines indicate patients with opposite patterns of serum creatinine changes on days 2–3. Day 2 (**E**) and day 3 (**F**) KeGFR values in subgroups of AP patients with serum creatinine (sCr) increase by ≥26.5 µmol/L or decrease by ≥26.5 µmol/L over the first three days of hospital stay. Data are shown as median (central line), interquartile range (box), non-outlier range (whiskers) and raw data (points).

**Figure 4 jcm-11-06159-f004:**
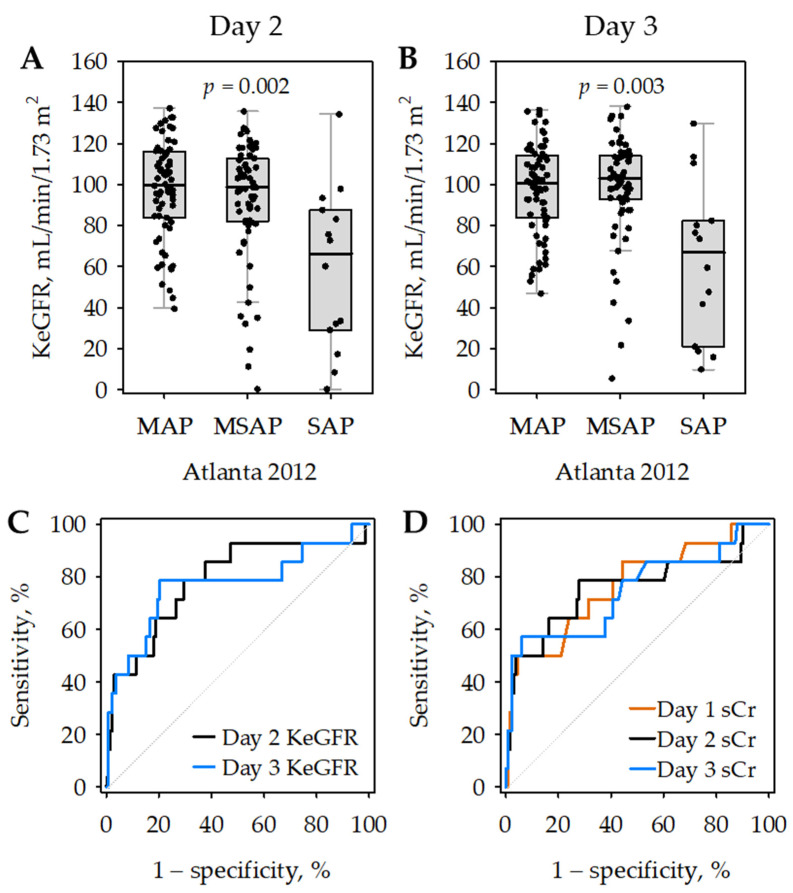
Kinetic estimated glomerular filtration rate (KeGFR) values on day 2 (**A**) and day 3 (**B**) of hospital stay according to the severity of acute pancreatitis (AP); *p*-values were calculated in Kruskal–Wallis test; patients with severe AP (SAP) differed significantly from those with mild AP (MAP) and moderately severe AP (MSAP) in post hoc tests. Data are shown as median (central line), interquartile range (box), non-outlier range (whiskers) and raw data (points). Receiver operating characteristic (ROC) curves showing the diagnostic accuracy of day 2 (black) and day 3 (blue) KeGFR for SAP (**C**). For comparison, ROC curves for serum creatinine (sCr) as a predictor of SAP are presented on panel (**D**) (day 1—orange; day 2—black; day 3—blue).

**Table 1 jcm-11-06159-t001:** Clinical characteristics of patients with AP included in the study.

Characteristic	MAP (*n* = 67)	MSAP (*n* = 66)	SAP (*n* = 14)	*p*-Value
Median age (Q1; Q3), years	56 (36; 73)	56 (41; 67)	60 (38; 77)	0.7
Female sex, *n* (%)	30 (45)	24 (36)	4 (29)	0.4
Preexisting comorbidity, *n* (%)	39 (58)	35 (53)	11 (79)	0.2
Diabetes, *n* (%)	5 (7)	9 (14)	3 (21)	0.3
Pulmonary disease, *n* (%)	4 (6)	4 (6)	0	0.6
Chronic kidney disease, *n* (%)	0	4 (6)	1 (7)	0.1
Ischemic heart disease, *n* (%)	14 (21)	20 (30)	9 (64)	0.005 ^a,b^
BMI > 30 kg/m^2^, *n* (%)	6 (9)	15 (23)	3 (21)	0.09
Etiology:				0.002 ^c^
Biliary, *n* (%)	33 (49)	23 (35)	3 (21)	
Alcohol, *n* (%)	19 (28)	12 (18)	8 (57)	
Hypertriglyceridemia, *n* (%)	7 (10)	3 (5)	0	
Idiopathic, *n* (%)	8 (12)	26 (39)	3 (21)	
Other, *n* (%)	0	2 (3)	0	
Median BISAP (Q1; Q3), points	1 (0; 2)	2 (2; 3)	3 (2; 4)	<0.001 ^a,b,c^
BISAP ≥ 3 points, *n* (%)	1 (1)	17 (26)	10 (71)	<0.001 ^a,b,c^
SIRS, *n* (%)	17 (25)	51 (77)	13 (93)	<0.001 ^a^
Pleural effusion, *n* (%)	21 (31)	60 (91)	13 (93)	<0.001 ^a,c^
Pancreatic or peripancreatic necrosis, *n* (%)	0	10 (15)	7 (50)	<0.001 ^a,b,c^
Organ failure:				<0.001 ^a,b,c^
Transient, *n* (%)	0	53 (80)	0	
Persistent, *n* (%)	0	0	14 (100)	
Renal failure (MMSS), *n* (%)	0	5 (8)	7 (50)	<0.001 ^a,b,c^
ERCP, *n* (%)	1 (1)	6 (9)	2 (14)	0.08
Surgery, *n* (%)	0	6 (9)	4 (29)	<0.001 ^a,c^
Parenteral nutrition, *n* (%)	0	4 (6)	8 (57)	<0.001 ^a,b^
Median length of hospital stay (Q1; Q3), days	6 (5; 8)	12 (10; 15)	26 (13; 31)	<0.001 ^a,c^
Mortality, *n* (%)	0	2 (3)	5 (36)	<0.001 ^a,b^

Significant differences in post hoc tests are denoted by superscript letters: ^a^—MAP vs. SAP; ^b^—MSAP vs. SAP; ^c^—MAP vs. MSAP. Abbreviations: MAP, mild acute pancreatitis; MSAP, moderately severe acute pancreatitis; SAP, severe acute pancreatitis; *n*, number of patients; Q1, lower quartile; Q3, upper quartile; BMI, body mass index; BISAP, Bedside Index for Severity in Acute Pancreatitis; SIRS, systemic inflammatory response syndrome; MMSS, modified Marshall scoring system; ERCP, endoscopic retrograde cholangiopancreatography.

**Table 2 jcm-11-06159-t002:** The results of laboratory tests on admission (day 1). Data are presented as median (lower; upper quartile) or mean ± standard deviation.

Laboratory Test	MAP (*n* = 67)	MSAP (*n* = 66)	SAP (*n* = 14)	*p*-Value
White blood cell count, ×10^3^/µL	11.6 (9.2; 14.6)	13.1 (10.5; 16.2)	13.7 (9.8; 21.9)	0.1
Hematocrit, %	42.6 ± 4.2	42.7 ± 5.7	43.9 ± 7.3	0.7
C-reactive protein, mg/L	11.2 (2.7; 67.2)	27.1 (12.9; 170.5)	132.8 (31.4; 287.4)	<0.001 ^a,c^
Amylase, U/L	847 (432; 1648)	791 (248; 2222)	585 (357; 1227)	0.9
Glucose, mmol/L	7.41 (5.74; 9.57)	8.42 (6.78; 9.33)	9.33 (6.94; 13.00)	0.033 ^a^
D-dimer, µg/mL	1.54 (0.88; 2.30)	2.14 (1.27; 3.74)	2.82 (2.10; 13.41)	0.003 ^a^
Albumin, g/L	40.6 (37.7; 44.0)	37.0 (34.0; 40.0)	35.2 (28.1; 39.0)	<0.001 ^a,c^
Urea, mmol/L	4.90 (3.34; 6.58)	5.00 (3.58; 6.60)	9.83 (6.67; 13.45)	<0.001 ^a,b^
Creatinine, µmol/L	71.2 (61.6; 89.3)	72.5 (60.3; 91.8)	106.4 (76.0; 191.8)	0.005 ^a,b^
Urea/creatinine ratio	70.3 (50.6; 85.9)	67.6 (50.4; 84.6)	77.8 (70.9; 96.3)	0.2
Cystatin C, mg/L *	0.87 (0.67; 1.07)	0.84 (0.77; 1.18)	1.66 (1.27; 2.42)	0.010 ^a,b^
BTP, mg/L *	0.45 (0.28; 0.67)	0.48 (0.21; 0.55)	1.06 (0.38; 1.17)	0.09
Uromodulin, ng/mL *	180 (117; 232)	162 (93; 201)	104 (79; 158)	0.2
Urine NGAL, ng/mL *	23.3 (14.2; 33.8)	72.8 (57.0; 380.5)	689.0 (408.7; 856.2)	<0.001 ^a,c^
Serum NGAL, ng/mL *	104 (68; 139)	181 (116; 274)	329 (199; 427)	0.007 ^a^
Angiopoietin-2, ng/mL *	2.96 (2.05; 4.02)	3.19 (2.29; 4.22)	9.71 (6.62; 20.56)	<0.001 ^a,b^
sFlt-1, pg/mL *	128 (106; 154)	144 (120; 172)	190 (182; 232)	0.001 ^a^

* Data available in subgroups of patients as specified in Appendix A (Table A1). Significant differences in post hoc tests are denoted by superscript letters: ^a^—MAP vs. SAP; ^b^—MSAP vs. SAP; ^c^—MAP vs. MSAP. Abbreviations: see Table 1; BTP, β-trace protein; NGAL, neutrophil gelatinase-associated protein; sFlt-1, soluble fms-like tyrosine kinase 1.

**Table 3 jcm-11-06159-t003:** Correlations between kinetic estimated glomerular filtration rate (KeGFR) values on day 2 and day 3 of hospital stay and the selected markers of AP severity and kidney function.

Variable	Day 2		Day 3	
	R	*p*-Value	R	*p*-Value
Age	−0.77	<0.001	−0.81	<0.001
BISAP	−0.28	<0.001	−0.26	0.001
Urea	−0.56	<0.001	−0.57	<0.001
Creatinine	−0.68	<0.001	−0.63	<0.001
Urea/creatinine ratio	−0.16	0.049	−0.19	0.020
Cystatin C	−0.73	<0.001	−0.88	<0.001
BTP	−0.62	<0.001	−0.86	<0.001
Uromodulin	0.34	0.014	0.29	0.037
Urine NGAL	−0.23	0.1	−0.10	0.5
Serum NGAL	−0.26	0.040	−0.25	0.041
Angiopoietin-2	−0.43	<0.001	−0.25	0.043
sFlt-1	−0.26	0.009	not assessed	

R—Spearman correlation coefficient. Abbreviations: see Table 1 and Table 2.

**Table 4 jcm-11-06159-t004:** Diagnostic accuracy of day 2 and day 3 KeGFR in diagnosis of renal failure defined according to the modified Marshall scoring system.

Variable	AUROC (95% CI)	Cut-off Value, mL/min/1.73 m^2^	Se, %	Sp, %	LR+	LR−
Day 2 KeGFR	0.942 (0.883–1.000)	≤35.5	75.0	97.8	33.75	0.26
		≤87.4	100	71.9	3.55	0
Day 3 KeGFR	0.950 (0.899–1.000)	≤47.9	75.0	98.5	50.62	0.25
		≤86.2	100	77.8	4.50	0

Abbreviations: AUROC, area under the ROC curve; CI, confidence interval; Se, diagnostic sensitivity; Sp, diagnostic specificity; LR+, likelihood ratio for positive result; LR−, likelihood ratio for negative result; KeGFR, kinetic estimated glomerular filtration rate.

**Table 5 jcm-11-06159-t005:** Multiple logistic regression model to predict renal failure defined according to the modified Marshall scoring system. Adjusted odds ratios (OR) are reported per 1 unit increase in the value of independent variable.

Independent Variable	Model 1		Model 2	
	OR (95% CI)	*p*-Value	OR (95% CI)	*p*-Value
Day 2 KeGFR, mL/min/1.73 m^2^	0.88 (0.80–0.96)	0.006	not included	
Day 3 KeGFR, mL/min/1.73 m^2^	not included		0.83 (0.73–0.94)	0.004
Age, years	0.94 (0.86–1.03)	0.2	0.83 (0.71–0.98)	0.023
Female sex	0.11 (0.004–2.93)	0.2	0.03 (0.0005–2.36)	0.1
Presence of comorbidities	2071 (1.08–4.0 × 10^6^)	0.046	207 (0.25–1.7 × 10^5^)	0.1
Diagnosis of SAP	7.15 (0.49–104)	0.1	0.93 (0.03–25.1)	0.9
Wadowice study center	17.1 (0.46–629)	0.1	4.22 (0.09–199)	0.5

Abbreviations: see Table 4.

**Table 6 jcm-11-06159-t006:** Diagnostic accuracy of day 2 and day 3 KeGFR in prediction of severe acute pancreatitis (SAP).

Variable	AUROC (95% CI)	Cut-Off Value, mL/min/1.73 m^2^	Se, %	Sp, %	LR+	LR−
Day 2 KeGFR	0.788 (0.647–0.928)	≤87.4	78.6	70.7	2.68	0.30
Day 3 KeGFR	0.769 (0.609–0.930)	≤82.4	78.6	79.7	3.87	0.27

Abbreviations: see Table 4.

**Table 7 jcm-11-06159-t007:** Multiple logistic regression model to predict severe acute pancreatitis (SAP). Adjusted odds ratios (OR) are reported per 1 unit increase in the value of independent variable.

Independent Variable	Model 1		Model 2	
	OR (95% CI)	*p*-Value	OR (95% CI)	*p*-Value
Day 2 KeGFR, mL/min/1.73 m^2^	0.96 (0.94–0.98)	<0.001	not included	
Day 3 KeGFR, mL/min/1.73 m^2^	not included		0.95 (0.92–0.97)	<0.001
Age, years	0.97 (0.93–1.02)	0.2	0.95 (0.90–1.002)	0.058
Female sex	0.53 (0.12–2.27)	0.4	0.50 (0.11–2.20)	0.3
Presence of comorbidities	7.60 (1.04–55.57)	0.044	4.81 (0.75–30.73)	0.09
Wadowice study center	4.03 (0.86–18.92)	0.07	2.70 (0.59–12.38)	0.2

Abbreviations: see Table 4.

## Data Availability

The data are available from the corresponding author upon reasonable request.

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
