# Peer review of "Acute Changes in Serum Creatinine and Kinetic Glomerular Filtration Rate Estimation in Early Phase of Acute Pancreatitis"

_jcm, 2022, doi:10.3390/jcm11206159_

Round 1

Reviewer 1 Report

1.  In the “Introduction” part, it is suggested to briefly introduce the research status of KeGFR.

2.  In table1, patients with prior CKD were also included. Whether this would affect the accuracy of the results of this study, it is recommended to exclude such patients.

3.  Page 5 line 219, the authors mention that “renal failure was diagnosed in 12 (8%) patients”. Does this indicate that there are not many patients with AP complicated with renal failure? Is it still necessary to use KeGFR to judge the renal function of AP?

4.  Page 6 line 229, why should you compare the different conditions of patients in the two centers, and what is the significance? If there is no realistic guiding significance, it is recommended to delete.

5.  AP may be multiple organ dysfunction. Is it not rigorous to judge its severity only by KeGFR?

6.  The article does not explain the relevant charts in many places, such as figure 3 B, D. So, it is suggested to explain the relevant charts in the results section.

7.  What is the KeGFR value using to determine the AP severity?

8.The sample size of this study was too small.

1、  In the “Introduction” part, it is suggested to briefly introduce the research status of KeGFR.

2、  In table1, patients with prior CKD were also included. Whether this would affect the accuracy of the results of this study, it is recommended to exclude such patients.

3、  Page 5 line 219, the authors mention that “renal failure was diagnosed in 12 (8%) patients”. Does this indicate that there are not many patients with AP complicated with renal failure? Is it still necessary to use KeGFR to judge the renal function of AP?

4、  Page 6 line 229, why should you compare the different conditions of patients in the two centers, and what is the significance? If there is no realistic guiding significance, it is recommended to delete.

5、  AP may be multiple organ dysfunction. Is it not rigorous to judge its severity only by KeGFR?

6、  The article does not explain the relevant charts in many places, such as figure 3 B, D. So, it is suggested to explain the relevant charts in the results section.

7、  What is the KeGFR value using to determine the AP severity?

Author Response

Manuscript no: jcm-1927562

Manuscript title: Acute Changes in Serum Creatinine and Kinetic Glomerular Filtration Rate Estimation in Early Phase of Acute Pancreatitis

Response to Reviewer 1

The authors thank the Reviewer for the interest in our manuscript and a comprehensive evaluation of its content. We highly appreciate the suggestions. We have carefully read and addressed the comments of both Reviewers and thoroughly revised the manuscript. Below, we present the detailed description of the changes introduced upon the revision and the explanations in response to the questions of the Reviewer. We hope that the revision together with our explanations will meet the requirements of the Reviewer and the Editor.

  1. In the “Introduction” part, it is suggested to briefly introduce the research status of KeGFR.

Response: Thank you for the suggestion. The respective paragraph in the Introduction (lines 79-93) has been supplemented with more information on recent studies assessing KeGFR in various patients’ populations.

  1. In table1, patients with prior CKD were also included. Whether this would affect the accuracy of the results of this study, it is recommended to exclude such patients.

Response: Thank you for pointing us to this important issue. Indeed, the study did not include patients with a medical record of chronic kidney disease in stage 3b-5, with known baseline serum creatinine concentrations above 170 µmol/L. This enabled us to literally use of modified Marshall scoring system to diagnose kidney failure during acute pancreatitis. We have added the explanations in Methods, lines 123-124 and Results, lines 229-232. We did not decide to exclude patients with less advanced chronic kidney disease because the disease is common, underdiagnosed, and is a known risk factor for acute kidney injury. Moreover, when chronic kidney disease was included as a confounder in our multiple logistic regression models (instead of all comorbidities), the results were very similar (please see the added information in lines 317-320).

  1. Page 5 line 219, the authors mention that “renal failure was diagnosed in 12 (8%) patients”. Does this indicate that there are not many patients with AP complicated with renal failure? Is it still necessary to use KeGFR to judge the renal function of AP?

Response: Although renal failure was diagnosed in 8% of our patients, acute changes in serum creatinine were much more common. In our study, renal failure was diagnosed according to modified Marshall scoring system, i.e. when serum creatinine exceeded 170 µmol/L (Methods, line 130), which is in line with 2012 Atlanta classification of the severity of acute pancreatitis. This definition is not consistent with the KDIGO definition of acute renal injury (AKI), however, the advantage of Marshall score for clinical practice is the use of single creatinine measurement instead of repeated measurements in KDIGO guidelines. In the paragraph starting in line 338, we report that the changes in serum creatinine exceeding the KDIGO diagnostic threshold for AKI of 26.5 µmol/L were observed in 23% of studied patients. Please note that this is almost twice the number of patients diagnosed with renal failure according to Marshall score, and forms a substantial proportion of patients.

  1. Page 6 line 229, why should you compare the different conditions of patients in the two centers, and what is the significance? If there is no realistic guiding significance, it is recommended to delete.

Response: Although the inclusion/exclusion criteria were the same in the two centers, the severity of AP and the age of patients differed between the centers. We presented the data in Results to justify our decision of including study center as a cofounder in multiple analyses. However, we agree with the Reviewer that this paragraph may be unnecessary in regard to KeGFR assessment. We deleted the paragraph in line with the Reviewer’s suggestion.

  1. AP may be multiple organ dysfunction. Is it not rigorous to judge its severity only by KeGFR?

Response: In our study, the severity of acute pancreatitis was diagnosed according to the 2012 Atlanta classification, i.e. based on transient or persistent cardiovascular, respiratory and kidney failure, local complications, and systemic complications (exacerbation of preexisting comorbidities) occurring during the course of AP. We evaluated KeGFR as a predictor of AP severity: we observed that low KeGFR on days 2 and 3 of hospital stay was associated with subsequent diagnosis of the severe AP and with AP mortality. In this part, the design of our study is similar to many previous studies evaluating single or multiple laboratory markers as early predictors of AP severity (e.g. doi: 10.1038/ajg.2015.370).  

  1. The article does not explain the relevant charts in many places, such as figure 3 B, D. So, it is suggested to explain the relevant charts in the results section.

Response: As suggested, we have added an explanation in Results, lines 352-353.

  1. What is the KeGFR value using to determine the AP severity?

Response: We did not mean to propose KeGFR as a diagnostic marker for severe AP. What we have shown is only that low KeGFR observed on initial days predicts the severe course of AP and mortality. Table 6 (line 369) shows the cut-off values of day 2 and 3 KeGFR for the prediction of severe AP and related diagnostic sensitivity and specificity.

  1. The sample size of this study was too small.

Response: We agree with the reviewer that a higher number of patients would allow more detailed analyses. We acknowledged the sample size as a limitation of the study (lines 563-564).

Reviewer 2 Report

The authors calculated estimate glomerular filtration rate (eGFR) in patients with acute pancreatitis (AP) using the kinetic eGFR (KeGFR) formula presented by Chen [20] and evaluated the diagnostic utility of KeGFR in the early stage of AP to assess renal function and to predict severity of AP.

As authors mentioned in conclusion session, there were needed to compare various ways of kinetic GFR estimation various equations and various filtration markers with measured GFR in patients with acutely changing kidney function to reveal suitable and accurate equations for AP. Authors cited some references, they filled their lacking data and described their opinions by using relatively recent references.

In materials and methods section, authors described as two KeGFR values during the study: on day 1 (based on serum creatinine measured on admission and on day 1), and day 2 (based on day 1 and day 2 serum creatinine results) (L. 188-190).

However, in the results section, authors described two KeGFR values were calculated in each patient during the study period: day 2 KeGFR was estimated based on serum creatinine concentrations measured on admission and on day 2 of hospital stay, while day 3 KeGFR was based on serum creatinine concentrations measured on days 2 and 3 (table 3 and L. 250-253).

Although KeGFR of day 1 that describes in materials and methods section, there was not describes in results section.

This description may confuse the reader.

Author Response

Manuscript no: jcm-1927562

Manuscript title: Acute Changes in Serum Creatinine and Kinetic Glomerular Filtration Rate Estimation in Early Phase of Acute Pancreatitis

Response to Reviewer 2

We are grateful for the interest in our manuscript and evaluation of its content. We have carefully read the comments of both Reviewers and thoroughly revised our manuscript. Below, we present the description of the changes introduced upon the revision and the explanations. We hope that the revision together with our explanations will meet the requirements of the Reviewer and the Editor.

The authors calculated estimate glomerular filtration rate (eGFR) in patients with acute pancreatitis (AP) using the kinetic eGFR (KeGFR) formula presented by Chen [20] and evaluated the diagnostic utility of KeGFR in the early stage of AP to assess renal function and to predict severity of AP.

As authors mentioned in conclusion session, there were needed to compare various ways of kinetic GFR estimation various equations and various filtration markers with measured GFR in patients with acutely changing kidney function to reveal suitable and accurate equations for AP. Authors cited some references, they filled their lacking data and described their opinions by using relatively recent references.

Response: Thank you for the comments.

In materials and methods section, authors described as two KeGFR values during the study: on day 1 (based on serum creatinine measured on admission and on day 1), and day 2 (based on day 1 and day 2 serum creatinine results) (L. 188-190).

However, in the results section, authors described two KeGFR values were calculated in each patient during the study period: day 2 KeGFR was estimated based on serum creatinine concentrations measured on admission and on day 2 of hospital stay, while day 3 KeGFR was based on serum creatinine concentrations measured on days 2 and 3 (table 3 and L. 250-253).

Although KeGFR of day 1 that describes in materials and methods section, there was not describes in results section.

This description may confuse the reader.

Response: We are grateful for spotting the error and we apologize for the inconsistency. In fact, serum creatinine was assessed three times during the study: on admission (= day 1), and on the following two days (= day 2 and day 3). As reported in Results, we calculated day 2 KeGFR based on day 1 and day 2 results, and day 3 KeGFR based on day 2 and day 3 results.

The respective information in Methods has been corrected (lines 197-198).

In response to another Reviewer’s suggestion, we have added more details regarding the included patients with chronic kidney disease. Briefly, the study included five patients with stage 1-3a chronic kidney disease (patients with more advanced CKD have been excluded), and two of them developed renal failure diagnosed according to modified Marshall scoring system. In multiple logistic regression, KeGFR was diagnostic for renal failure independently of preexisting chronic kidney disease.